# Electrochemotherapy vs radiotherapy in the treatment of primary cutaneous malignancies or cutaneous metastases from primary solid organ malignancies: A systematic review and narrative synthesis

**Angus Torry McMillan**[ID]1*, **Luke McElroy**2, **Lorcan O'Toole**3, **Paolo Matteucci**1, **Joshua Philip Totty**[ID]1,4,5*

1 Department of Plastic and Reconstructive Surgery, Hull University Teaching Hospitals NHS Trust, Cottingham, United Kingdom, 2 Department of General Surgery, Forth Valley Royal Hospital, Larbert, United Kingdom, 3 Department of Oncology, Hull University Teaching Hospitals NHS Trust, Cottingham, United Kingdom, 4 Centre for Clinical Sciences, Hull York Medical School, Hull, United Kingdom, 5 University of Hull, Cottingham Road, Hull, United Kingdom

* angusmcmillan@doctors.net.uk (ATM); joshua.totty@hyms.ac.uk (JPT)

## Abstract

### Background

Electrochemotherapy has gained international traction and commendation in national guidelines as an effective tool in the management of cutaneous malignancies not amenable to surgical resection. Despite this, no level 5 evidence exists comparing it to radiotherapy in the treatment of cutaneous malignancies. This systematic review aimed to examine the literature directly and indirectly comparing electrochemotherapy and radiotherapy in the treatment of primary cutaneous malignancies or cutaneous metastases from primary solid organ malignancies.

### Materials & methods

The protocol for this review was registered on the PROSPERO International Prospective Register of Systematic Reviews with the protocol ID CRD42021285415. Searches of MEDLINE, Embase, CINAHL, CENTRAL and ClinicalTrials.gov databases were undertaken from database inception to 28 December 2021. Studies in humans comparing treatment with electrochemotherapy to radiotherapy and reporting tumour response with a minimum four week follow-up were eligible. Risk of bias was assessed using the ROBINS-I tool. Results are provided as a narrative synthesis.

### Results

Two case series with a total of 92 patients were identified as relevant to this study. Both case series examined patients with cutaneous squamous cell carcinoma. One case series examined elderly patients with predominantly head/neck lesions. The other examined

**Data Availability Statement:** All relevant data are within the paper and its Supporting Information files.

**Funding:** The authors received no specific funding for this work. Joshua P Totty is a Clinical Lecturer (CL-2020-03-001) funded by Health Education England (HEE) / National Institute for Health Research (NIHR). The views expressed in this publication are those of the author(s) and not necessarily those of the NIHR, NHS or the UK Department of Health and Social Care. Neither the NIHR, NHS, or the UK Department of Health and Social Care had any role in study design, data collection and analysis, decision to publish, or preparation of the manuscript.

**Competing interests:** The authors have declared that no competing interests exist.

younger patients with predominantly limb lesions who had cutaneous squamous cell carcinoma directly attributable to a rare skin condition.

## Conclusion

There is little literature presenting comparative data for electrochemotherapy and radiotherapy in the treatment of primary cutaneous malignancies or cutaneous metastases. Included studies were marred by serious risk of bias particularly due to confounding. The inherent bias and heterogeneity of the included studies precluded synthesis of a consolidated comparison of clinical outcomes between the two therapies. Further research is required in this domain in the form of clinical trials and observational studies to inform guidelines for electrochemotherapy treatment.

## Introduction

Primary skin malignancies are a global public health threat with a significant impact on mortality and morbidity, particularly amongst the older population [1]. Surgical resection with or without reconstruction remains the gold-standard curative treatment option [2,3]. The incidence of skin malignancies is higher in the clinically frail patient, in whom elevated rates of postoperative complications are seen [4,5]. In some patients, there are factors which preclude excision as the treatment of choice. Tumour size, location, plurality and co-morbidity all influence the feasibility of curative surgical intervention. Radiotherapy is a well-established treatment modality which can be offered with palliative intent in the management of a number of cutaneous carcinomas and sarcomas [6–8]. Electrochemotherapy is an alternative treatment which has been gaining international traction in specialist centres since the publication of standard operating procedures for its use [9,10]. The treatment relies on the principle of electroporation—the use of electrical current to increase the permeability of cells to cytotoxic agents—and has been shown to have greater efficacy than the administration of chemotherapy agents alone [11]. Previous systematic reviews have demonstrated the efficacy of electrochemotherapy in the treatment of primary cutaneous malignancies [11–13], and in the treatment of cutaneous metastases from other primary solid organ malignancies [14–16]. The National Institution for Health and Clinical Excellence have issued guidance for the use of electrochemotherapy in specialist settings, though treatment selection is still deferred to the opinion of the loco-regional multi-disciplinary team [17,18].

A systematic review and meta-analysis by Spratt et al. examined non-comparative trials to estimate the efficacy of different skin-directed therapies in the treatment of cutaneous metastases [14]. They synthesised independent estimates of the efficacy of electrochemotherapy and radiotherapy, inferring conclusions from low-level evidence. To date, there has been no level 5 evidence evaluating directly or indirectly comparative studies of electrochemotherapy and radiotherapy in the treatment of primary cutaneous malignancies or cutaneous metastases from primary solid organ malignancies.

The objective of this study, therefore, was to systematically examine the published literature directly and indirectly comparing electrochemotherapy and radiotherapy in the treatment of patients with primary cutaneous malignancies unsuitable for curative surgical resection or cutaneous metastases from primary solid organ malignancies.

## Materials and methods

The protocol for this review was registered on the PROSPERO International Prospective Register of Systematic Reviews [19] with the protocol ID CRD42021285415 and has been published in a peer-reviewed journal [20]. A copy of the protocol registered on PROSPERO is provided in the supporting information. The review has been conducted in line with the Cochrane Handbook for Systematic Reviews of Interventions [21] and the manuscript prepared in accordance with the Preferred Reporting Items for Systematic Reviews and Meta-Analyses (PRISMA) statement and checklist [22]. A completed PRISMA flow diagram is provided in the supporting information. Ethical approval was not required for this study as it extracted data from previous studies in which informed consent had been obtained by the primary researches.

### Eligibility

Studies were considered eligible for inclusion if they compared treatment with electrochemotherapy to treatment with radiotherapy and reported on tumour response after treatment delivery with at least a four-week follow-up period. Studies were deemed suitable for inclusion in a meta-analysis if they presented comparable data for tumour response between electrochemotherapy and radiotherapy treatment groups and were of suitable clinical homogeneity.

Only studies applying to humans were included. Studies were included regardless of their publication language or country of origin.

### Search sources and strategy

A comprehensive search strategy was developed to encompass all relevant works relating to the review. With the help of an information specialist, appropriate MeSH and free-text search term were identified and combined with Boolean operators. Using a search strategy designed with an information specialist, the Healthcare Databases Advanced Search tool was used to search the databases MEDLINE, Embase, and CINAHL from the time period from database inception to 28 December 2021. The CENTRAL and ClinicalTrials.gov registries were also searched. Supplementary searches of grey literature were undertaken via Web of Science, SCOPUS and Zetoc. The full search strategy for each source is provided in the supporting information. Review and guideline articles identified by the search, in addition to retrieved articles, underwent manual bibliography searching to identify studies missed by searches. Any additional studies that were suggested by authors who were approached for full texts were also screened.

Results from the searches were combined and uploaded to Rayyan, an open source tool designed for systematic reviews [23], and deduplicated. Titles and abstracts were independently screened by two authors (AM and LM) against the inclusion/exclusion criteria. Any disagreement was moderated by a senior author (JT), who made a final decision on inclusion. Following title and abstract screening, articles were retrieved, and full text screening was undertaken by two authors (AM and LM) acting independently. Any disagreement was again moderated by a senior author (JT) with disagreements resolved through discussion.

### Study selection

All studies comparing electrochemotherapy and radiotherapy in the treatment of primary cutaneous malignancies or cutaneous metastases from other primary solid organ malignancies were eligible for inclusion. Criteria for study selection were defined using the Population,

Intervention, Comparison, Outcome (PICO) framework outlined in the pre-published protocol [20].

### Study design

Randomised control trials, cohort studies, case control studies and case series were all eligible for inclusion in the review. Case series were not eligible for inclusion in any meta-analysis. There were no limitations made based upon patient selection criteria or study size. Letters, opinion pieces, literature reviews, and case reports were excluded.

### Data extraction

Two authors (AM and LM) individually extracted the data and recorded it in a pre-designed electronic form. The two authors compared collected data and if a consensus could not be reached, a third author (JT) resolved any disagreement.

The following data was collected for comparison:

- Study characteristics, funding source, patient demographics, response evaluation time, recruitment/sampling procedures and tumour volume response evaluation method

- Tumour anatomy, number, size and histotype

- Electrochemotherapy agent, route and operating procedure technique

- Radiotherapy technique and characteristics

- Tumour volume response, which was the primary outcome of this study, and any secondary outcomes reported by the study such as patient reported outcome measures, pain, toxicity/ adverse events and progression-free survival

If any of these data could not be directly extracted from a study, the authors of the study were contacted by email for further information.

### Risk of bias assessment

For each study included in the review which incorporated randomisation of participants, a risk of bias assessment was undertaken using the Cochrane Collaboration tool for assessing risk of bias in randomised trials [24]. For each study included in the review which did not randomise participants, a risk of bias assessment was undertaken using the ROBINS-I tool [25]. These tools were used to stratify studies into those at low, moderate, serious and critical risk of bias. Risk of bias was assessed independently by two authors (AM and LM) and the results were collated in a risk of bias assessment table.

### Differences from the protocol

In our pre-published protocol we planned to undertake a meta-analysis including sub-group analyses if there was suitable extraction of clinically homogenous data [20]. However due the paucity of the literature this was not possible. We had also planned to assess the quality of evidence of each study outcome using the Grading of Recommendations, Assessment, Development and Evaluation approach [26] and to assess for reporting bias using a funnel plot and a corresponding formal statistical test though this was also not possible due to a lack of suitable studies.

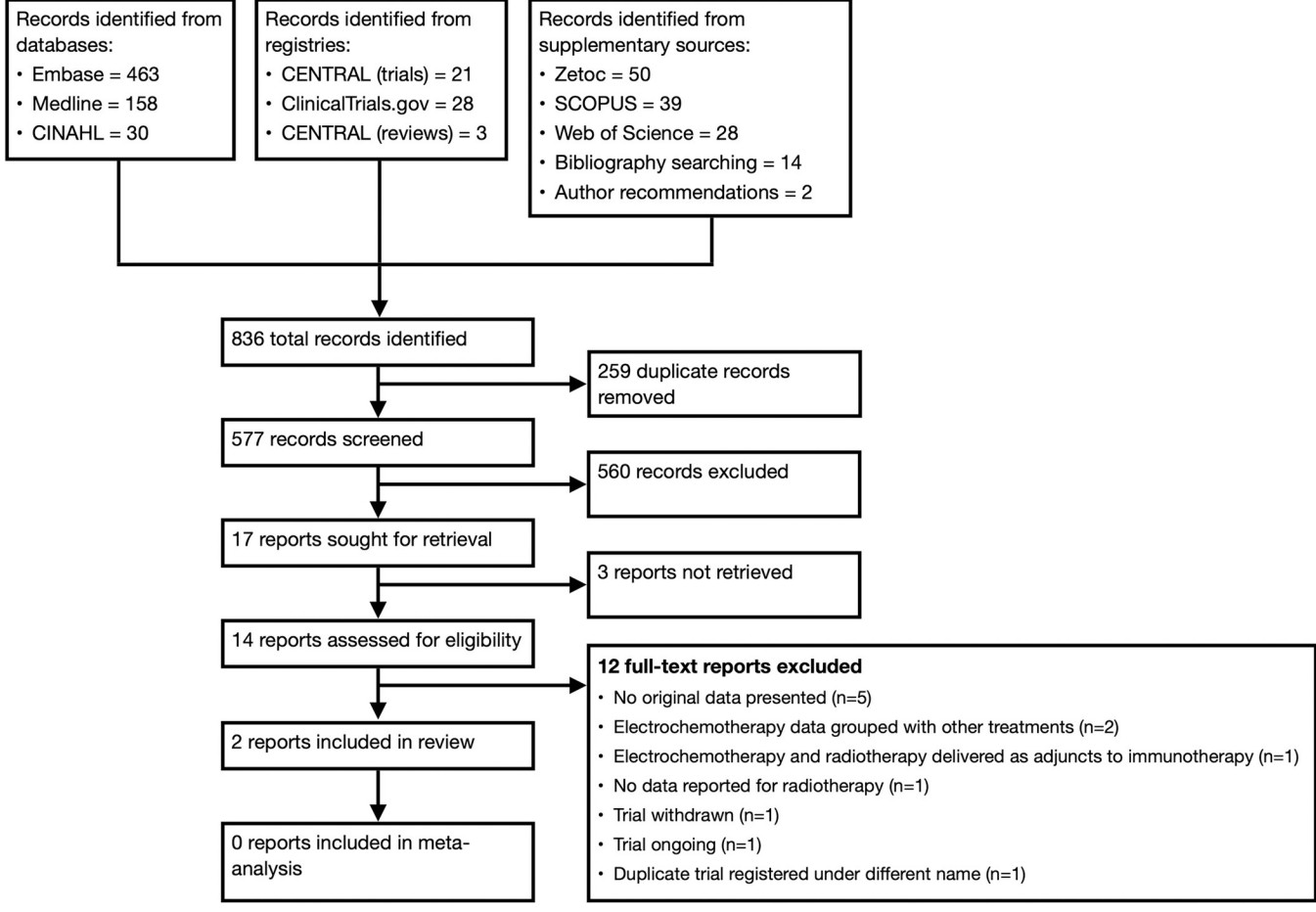

**Fig 1. The selection process for studies included in the review.**

## Results

In total 577 records were identified by the search strategy after removal of duplicates. After title and abstract screening, the full text for 17 records were sought for retrieval. The study selection procedure adapted from the PRISMA 2020 Statement is shown in Fig 1 [22]. After full-text screening, two studies were identified that compared electrochemotherapy and radiotherapy [27,28]. A summary of these can be seen in Table 1. There were insufficient studies for meta-analysis. Results are therefore provided as a narrative synthesis.

### Studies excluded and reasons for exclusion

Five studies were excluded after full-text screening due to a lack of original data. Two of these studies were systematic reviews which had been identified by the search [14,29], two were background articles about electrochemotherapy [30,31], and one was a paper about the cost efficacy of electrochemotherapy compared with other techniques which obtained its efficacy data from other studies [32].

Two studies were excluded as they grouped data for electrochemotherapy with other interventions such as radiotherapy as 'local therapies' and any outcomes were inseparable and not suitable for extraction [33,34]. We contacted the authors of these studies to obtain extractable data but did not receive a response.

**Table 1. A summary of included the studies.**

| Study Characteristics | Amaral et al. [27] | Robertson et al. [28] |
|---|---|---|
| *Study design* | Case series | Case series |
| *Study setting* | Single centre, Germany | Two centres, United Kingdom |
| *Study period* | January 2011 –June 2018 | July 1991 –June 2019 |
| *Study population* | Patients with advanced cutaneous squamous cell carcinoma (AJCC 2017 stage III and IV) | Patients with severe recessive dystrophic epidermolysis bullosa diagnosed with cutaneous squamous cell carcinoma |
| *Method of tumour response assessment* | Radiological assessment with CT imaging using RECIST criteria | Not stated |
| **Characteristics of patients treated with electrochemotherapy** | | |
| *Number of patients* | 3 | 2 |
| *Sex* | 3 (100%) male | 1 male (33.3%), 2 female (66.6%) |
| *Age at diagnosis* | 2 (66.7%) 71–80 yrs; 1 (33.3%) 81–90 yrs | 29.4 yrs (mean) |
| *Intervention characteristics* | Systemic bleomycin delivered to ESOPE guidelines [8] | Bleomycin (no further detail given) |
| *Tumour location* | 3 (100%) head/neck | Majority on limbs |
| *Tumour stage (AJCC 2017)* | 3 (100%) stage IV | Not stated |
| *Presence of metastases* | 2 (66.7%) locoregional metastases | 2 (66.7%) locoregional metastases |
| *Previous therapies* | 3 (100%) previous surgical treatment; 2 (66.7%) previous radiotherapy | 3 (100%) previous surgical treatment; 1 (33.3%) previous imiquimod and previous systemic retinoids |
| *Tumour response* | 2 (66.7%) progressive disease; 1 (33.3%) partial response | No significant tumour response |
| *Median progression free survival (IQR)* | 11 (8–174) months | Not stated |
| **Characteristics of patients treated with radiotherapy** | | |
| *Number of patients* | 82 | 4 |
| *Sex* | 61 (74.4%) male; 21 (25.6%) female | 1 (25%) male; 3 (75%) female |
| *Age at diagnosis* | 13 (16%) ≤70 yrs; 34 (41.5%) 71–80 yrs; 30 (36.6%) 81–90 yrs; 5 (6.1%) ≥90 yrs | 27.9 yrs (mean) |
| *Intervention characteristics* | Conventional non-stereotactic radiotherapy; 70 Gy delivered over 35 fractions | 45–50 Gy delivered over 20–25 fractions |
| *Tumour location* | 62 (75.6%) head/neck, 9 (10.9%) trunk, 7 (8.5%) limbs | Majority of lesions on limbs |
| *Tumour stage (AJCC 2017)* | 40 (48.8%) stage III; 42 (51.2%) stage IV | Not stated |
| *Presence of metastases* | 26 (31.7%) locoregional metastases; 2 distant metastases (2.4%) | 4 (100%) locoregional metastases |
| *Previous therapies* | 58 (70.7%) previous surgical treatment | 4 (100%) previous surgical treatment |
| *Tumour response* | Not stated | Slowed tumour growth |
| *Median progression free survival (IQR)* | 15 (8–45) months | Not stated |

Another study examining patients with melanoma had included a small number of patients treated with electrochemotherapy though no extractable data is given [35]. The authors mention the use of radiotherapy as an adjunct but do not provide any further detail.

A study by Guida et al. was excluded as it examined both electrochemotherapy and radiotherapy in the treatment of oligoprogressive metastatic melanoma in the context of patients who were also being treated with long-term immunotherapy, and this review focussed on monotherapy by either modality [36].

Two studies were identified on trial registries but excluded from this review. One of these trials was excluded due to the study ending before collecting any data [37]. This trial aimed to directly compare electrochemotherapy and radiotherapy in palliative treatment of ulcerated cutaneous metastases and was withdrawn due to lack of recruitment, with study contacts

outlining that the study did not accrue any patients due to practical difficulties at the hosting department. Additionally, this trial was duplicated on another registry under a similar name. The other study identified on trial registries was a randomised controlled trial comparing intratumoural bleomycin electrochemotherapy (delivered according to updated European standard operating procedure guidelines [10] with standard radiotherapy (51 Gy in 3 Gy fractions) in the treatment of basal cell carcinoma [38]. Unfortunately, this study was unsuitable for inclusion as it was still ongoing at the time of this review and unable to provide data [38].

## Risk of bias in included studies

The ROBINS-I tool was used to assess risk of bias in the included studies [25]. Overall risk of bias was judged to be serious for both studies. The risk of bias in each domain is summarised in Table 2. As the included studies are both case series, an important source of bias was bias due to confounding as patients were treated with different interventions according to factors related to their disease.

## Data extraction

The study by Amaral et al. was a retrospective case series that examined patients identified as having advanced cutaneous squamous cell carcinoma in a single centre between January 2011 and June 2018 [27]. Of 195 cases validated as advanced cutaneous squamous cell carcinoma, three were treated with electrochemotherapy and 82 were treated with radiotherapy. The original publication of this study did not contain any extractable data but the authors were contacted and provided further details on both radiotherapy and electrochemotherapy patients. All patients treated with electrochemotherapy and 58 patients treated with radiotherapy (80.1%) had surgery prior to treatment. Two of the patients treated with electrochemotherapy had also previously been treated with radiotherapy. All lesions treated with electrochemotherapy were head/neck (specifically scalp) lesions, whereas 62 (75.6%) of the lesions treated with radiotherapy were head/neck lesions, the remainder were present on the limbs/trunk.

Amaral et al. reported that of the three patients they treated with electrochemotherapy, two were evaluated as having progressive disease and one demonstrated a partial response [27]. Bleomycin was given via a systemic route for these patients and delivered in accordance to European standard operating procedure guidelines [10]. Conventional non-stereotactic radiotherapy was delivered at a dose of 70 Gy in single 2 Gy dose fractions to 82 patients. The authors did not collect any data regarding tumour response for these patients as this was not their primary endpoint. They did collect data on progression-survival (PFS) and patients in the radiotherapy group had a median PFS of 15 months (IQR 8–45 months) whereas patients in the electrochemotherapy group had a median PFS 11 months (8–174 months). The authors did not collect any data regarding pain, adverse events or other secondary outcomes.

Robertson et al. was a retrospective case series that examined all patients with epidermolysis bullosa who were diagnosed with squamous cell carcinoma and treated by Guy's and St

**Table 2. The risk of bias assessment for included the studies using the ROBINS-I tool.**

| Study | Bias due to confounding | Bias due to selection | Bias in classification of interventions | Bias due to deviations from intended interventions | Bias due to missing data | Bias in measurement of outcomes | Bias in selection of the reported result | Overall judgement |
|---|---|---|---|---|---|---|---|---|
| *Amaral et al.* [27] | Serious | Serious | Serious | Serious | No Information | Low | Moderate | Serious |
| *Robertson et al.* [28] | Serious | Serious | Serious | Serious | No information | No information | Moderate | Serious |

Thomas' NHS Foundation Trust and Great Ormond Street Hospital for Children NHS Foundation Trust over a 28 year period between 1 July 1991 and 30 June 2019 [28]. Of the 44 cases they reviewed, two cases of locally aggressive squamous cell carcinoma were treated with electrochemotherapy and four cases of ulcerated metastatic squamous cell carcinoma were treated with radiotherapy. All six of these cases had the severe recessive dystrophic subtype of epidermolysis bullosa. The locations of squamous cell carcinomae in each treatment group were not extractable from the paper though the authors stated the majority of squamous cell carcinomae were present on the limbs; none arose on the head or neck and few arose on the trunk.

Robertson et al. reported both patients who received electrochemotherapy demonstrated no tumour response [28]. The authors did not describe the tumour volume response evaluation method or response evaluation time. Bleomycin was the electrochemotherapy agent used but the route of administration and operating procedure was not described by the authors. Patients treated with electrochemotherapy reported significant pain and one developed severe post-operative sepsis. Radiotherapy was delivered with palliative intent at a dose of 45–50 Gy delivered over 20–25 fractions. The four patients treated with radiotherapy showed slowed tumour growth and improved wound healing. The authors were approached multiple times requesting missing data which could not be extracted, though no response was received.

## Discussion

Despite a thorough search strategy and robust methodology, our results were limited by the paucity of the literature. We identified just two studies which contained indirectly comparative data for electrochemotherapy and radiotherapy in the treatment of cutaneous malignancy [27,28]. These two case series contained only a small number of relevant patients treated with electrochemotherapy or radiotherapy. Whilst both studies examined patients with cutaneous squamous cell carcinoma, they were clinically heterogenous demographics. Amaral et al. examined patients who were predominantly elderly with advanced cutaneous squamous cell carcinoma lesions mostly presenting in the head/neck region [27], whereas the cases examined by Robertson et al. were relatively young patients with cutaneous squamous cell carcinoma lesions presenting mostly on the limbs and directly attributable to a rare skin condition (severe recessive dystrophic epidermolysis bullosa). Both case series were judged to be at serious risk of bias. One significant source of bias was through treatment allocation; in both studies, participants were allocated treatment based on disease-related factors. This is evidenced in the study by Robertson et al. where patients with locally aggressive disease were treated with electrochemotherapy whereas patients with ulcerated cutaneous metastases were treated with radiotherapy. Though individually, these two case series may be of interest to specialists involved in the treatment of either demographic, they are too clinically heterogenous to be meaningfully combined and their inherent risk of bias means these results are unlikely to be externally valid and should not be used to inform decision making about the use of electrochemotherapy or radiotherapy in the treatment of cutaneous malignancies in the general population.

Through searching, additional data sets which included groups of patients treated with electrochemotherapy and with radiotherapy were identified, but due to the how the outcomes for these patients were grouped, no data could be extracted to allow indirect comparison [33,34]. Notably, there was no published data from any directly comparative trials between electrochemotherapy and radiotherapy. Though we did identify a previously withdrawn and an ongoing randomised controlled trial comparing these two therapies [37,38]. This review highlights a marked deficit of comparative data between electrochemotherapy and radiotherapy in the published literature.

Previous systematic reviews of the literature have demonstrated that electrochemotherapy is a safe and effective tool in the treatment of metastases from primary solid organ

malignancies [14–16] and in the treatment of primary cutaneous malignances such as basal cell carcinoma [12] and melanoma [13]. A review by Mali et al. examined the effect of electrochemotherapy on multiple cutaneous and subcutaneous tumour types and found squamous cell carcinoma to have the lowest complete response rate (49.5%) and objective response rate (69.7%) in comparison to basal cell carcinoma which had the highest complete response rate (88.6%) and objective response rate (100.0%) of all tumour types [11]. The two studies identified by our review examined patients with squamous cell carcinomae treated with electrochemotherapy or radiotherapy [27,28]. In the small number of these patients who were treated with electrochemotherapy, the tumour response was generally not favourable. The literature suggests that electrochemotherapy may be less efficacious in this particular cohort of patients.

In their review of skin-directed therapies for cutaneous metastases, Spratt et al. estimated the complete response rate of electrochemotherapy to be 47.5% (95% CI 30.3%–65.3%) and the objective response rate to be 75.4% (95% CI 57.7%–87.3%) [14]. They also estimated the complete response rate for radiotherapy to be 62.7% (95% CI 22.8%–90.5%) and the objective response rate to be 83.8% (95% CI 37.9%–97.8%). These estimates were synthesised independently from non-comparative sources. Where this sits within the wider treatment landscape is unknown, and our review demonstrates there is little comparative data for these two treatment modalities.

The primary aim of this review was to compare electrochemotherapy with radiotherapy. Although there are other treatment modalities available for cutaneous malignancies and metastases from solid organ malignancies, such as systemic chemotherapy, immunotherapy, isolated limb perfusion, and intralesional therapies like interferon and interleukin-2, an extensive examination of the literature on these modalities was beyond the scope of this review. Each therapy has its distinct merits and flaws, which, in addition to clinical efficacy, need to be considered when determining the appropriate treatment for each patient. Therapies such as electrochemotherapy and isolated limb perfusion can induce a sustained response after just one or two treatment sessions a few weeks apart [39,40]. In contrast, radiotherapy and intralesional therapy require regular administration over several weeks [41,42]. Despite the time efficiency of electrochemotherapy and isolated limb perfusion, these therapies often necessitate the patient to undergo general anaesthesia in order to be tolerated [39,43]. In cases were patients are too high risk for general anaesthetia, radiotherapy, which can be delivered without anaesthesia, may be an suitable alternative [42]. Additionally, electrochemotherapy and radiotherapy have been recognised as highly cost-effective therapies, proving significantly more affordable than isolated limb perfusion and interferon treatment [32]. Moreover, the substantial costs associated with emerging treatments such as systemic immunotherapy, can limit their delivery to selected groups of patients [44,45]. Therefore, it is important not to disregard economical therapies such as electrochemotherapy and radiotherapy.

There is a clear need for further directly comparative data in the form of clinical trials (such as the ongoing trial in patients with basal cell carcinoma [38]) and observational studies to provide evidence for guidelines for the use of electrochemotherapy in the treatment of primary cutaneous malignancies and cutaneous metastases from other primary solid organ malignancies. Future clinical trials should not only investigate the comparative efficacy of these therapies but should also consider their convenience for patients, effect on quality of life, associated adverse events, and cost effectiveness. This comprehensive approach will allow clinicians to make well-informed decisions to tailor patient-specific management. Despite being limited by the paucity of literature, this review was undertaken with a robust methodology and should be repeated in the future when more directly comparative data is published.

## Conclusion

Despite electrochemotherapy showing promising efficacy in a variety of tumour types, there is an almost complete lack of literature containing comparative data for electrochemotherapy and radiotherapy in the treatment of primary cutaneous malignancies and metastases from other primary solid organ malignancies. The two studies identified by this review differ significantly in study population and outcomes, as such, their heterogeneity precludes synthesis of a consolidated comparison of electrochemotherapy and radiotherapy. Recommendations from NICE suggest it is suitable for use in select patient cohorts under the guidance of specialist skin cancer multi-disciplinary teams. Further research is required in this domain in the form of clinical trials and observational studies to inform patient selection guidelines for the use of electrochemotherapy.

## Supporting information

**S1 Checklist. The PRISMA 2020 checklist for the systematic review.**
(PDF)

**S1 Protocol. The protocol for the systematic review registered on PROSPERO.**
(PDF)

**S1 Search strategy. The full search strategy for the systematic review.**
(PDF)

## Acknowledgments

We would like to thank Tim Staniland, librarian at Hull University Teaching Hospitals NHS Trust, for his help with designing the search strategy.

## Author Contributions

**Conceptualization:** Angus Torry McMillan, Lorcan O'Toole, Paolo Matteucci, Joshua Philip Totty.

**Formal analysis:** Angus Torry McMillan.

**Investigation:** Angus Torry McMillan, Luke McElroy.

**Methodology:** Angus Torry McMillan, Joshua Philip Totty.

**Project administration:** Joshua Philip Totty.

**Supervision:** Lorcan O'Toole, Paolo Matteucci, Joshua Philip Totty.

**Validation:** Angus Torry McMillan, Luke McElroy, Joshua Philip Totty.

**Writing – original draft:** Angus Torry McMillan.

**Writing – review & editing:** Angus Torry McMillan, Luke McElroy, Lorcan O'Toole, Paolo Matteucci, Joshua Philip Totty.

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
