## [Decision Letter · Decision Letter 0]

24 Apr 2023

PONE-D-22-32656Electrochemotherapy vs radiotherapy in the treatment of primary cutaneous malignancies or cutaneous metastases from primary solid organ malignancies: A systematic review and narrative synthesisPLOS ONE

Dear Dr. Totty,

Thank you for submitting your manuscript to PLOS ONE. After careful consideration, we feel that it has merit but does not fully meet PLOS ONE’s publication criteria as it currently stands. Therefore, we invite you to submit a revised version of the manuscript that addresses the points raised during the review process. 

The manuscript has been evaluated by three reviewers, and their comments are available below. The reviewers have raised a number of concerns that need attention. They request additional information on methodological aspects of the study, as well as the interpretation and presentation of the results. Could you please revise the manuscript to carefully address the concerns raised?

We look forward to receiving your revised manuscript.

Kind regards,

Dario Ummarino, PhD

Senior Editor

PLOS ONE

Journal Requirements:

2. Please note that in order to use the direct billing option the corresponding author must be affiliated with the chosen institute. Please either amend your manuscript to change the affiliation or corresponding author, or email us at plosone@plos.org with a request to remove this option.

Reviewers' comments:

Reviewer's Responses to Questions

**Comments to the Author**

1. Is the manuscript technically sound, and do the data support the conclusions?

Reviewer #1: Yes

Reviewer #2: Partly

Reviewer #3: Yes

2. Has the statistical analysis been performed appropriately and rigorously? 

Reviewer #1: Yes

Reviewer #2: N/A

Reviewer #3: Yes

3. Have the authors made all data underlying the findings in their manuscript fully available?

Reviewer #1: Yes

Reviewer #2: Yes

Reviewer #3: Yes

4. Is the manuscript presented in an intelligible fashion and written in standard English?

Reviewer #1: Yes

Reviewer #2: Yes

Reviewer #3: Yes

5. Review Comments to the Author

Reviewer #1: I have read with great interest this systematic review about the role of electrochemotherapy for skin malignancies, in comparison with radiotherapy. The paper is well written and data are properly collected and presented. Correctly, the authors highlight the paucity of literature about the potential role of electrochemotherapy with very few experiences available. Nonetheless, I think that this article may help clinicians in the knowledge of this therapeutic option.

I only suggest to mention, in the Introduction, among the radiotherapy experiences for skin malignancies:

-doi: 10.1007/s00403-021-02268-1.

-doi: 10.4081/rt.2017.6942

Reviewer #2: McMillan et al. report a well-written systematic review of studies directly comparing electrochemotherapy and radiotherapy in the treatment of cutaneous solid tumour metastases. While this is an interesting topic, it is significantly limited by the lack of available literature for any meaningful data synthesis. Therefore, I do not believe this meets criteria for publication as such a lengthy article by PloS One. More detailed suggestions for revisions, if the authors decide to proceed to this, are made below.

- In Table 1 it would be useful for treating clinicians to know if there were any reported prior treatments to electrochemotherapy/RT to understand if these have a role in resistant or progressive disease. It would also be helpful to know how responses were measured in these studies (RECIST, radiological, clinically etc).

- The study by Amaral et al. reported on PFS outcomes with electrochemotherapy vs RT (line 253, page 13) - this should be included in the summary results table on page 12 as PFS is a clinically relevant outcome when deciding between treatment options

- Similarly, cohort details of the 2 provided studies should be provided in the summary table - as the fact that Robertson et al.'s study was limited to patients with epidermolysis bullosa was not mentioned until much later (line 258, page 13). This is relevant as these results may not be as readily generalisable to clinicians treating cutaneous SCC beyond this rare subgroup

- As acknowledged in the discussion, page 15 paragraph 1, the two studies are far too heterogenous both in design and in outcome reporting to allow for meaningful conclusions of electrochemotherapy vs radiotherapy to be drawn. This is a major issue with the review paper overall as it is not informative enough to guide practice, and this should be emphasised more clearly in the Abstract and conclusions.

- Paragraph 2, page 15 is a repetition from the Results, and does not need to be highlighted again

- Paragraph 3, page 15 is a description of studies identified in the literature search and should be moved to the Methods

- There needs to be more text in the Discussion describing the role of electrochemotherapy or radiotherapy in the current treatment landscape of cutaneous metastases/solid tumours - these are not the only treatment options for patients and by not mentioning alternatives (isolated limb perfusion/infusion, systemic medications including checkpoint inhibitors for cSCC/melanoma or HH inhibitors for BCC, intralesional therapies such as T-VEC or IL-2 etc) the manuscript implies clinicians are limited to these two options

- The practical benefits for patients in comparison to these alternatives should be discussed in more detail - e.g. reduced time/financial toxicities, reduced systemic side effects etc

- All in all, I do not think the narrow range of results and conclusions that can be drawn from this study justify publication as such as lengthy article. It would be more suitable to shorten it to a Letter or Correspondence format; or to expand upon the search strategy to include single-arm studies of either electrochemotherapy or radiotherapy in cutaneous metastases to provide a more meaningful indirect comparison (given that the two provided studies are both non-randomised and heterogenous themselves).

Thank you for the opportunity to review this manuscript and provide my suggestions for improvement, and I look forward to reviewing any future revisions.

Reviewer #3: Paper clear and easily understood even by the non-expert. Criteria for literature selection are thoroughly described. The conclusions are interlocutory, but this is due to lack of literature. I think it can be published.

6. PLOS authors have the option to publish the peer review history of their article (what does this mean?). If published, this will include your full peer review and any attached files.

Reviewer #1: No

Reviewer #2: No

Reviewer #3: **Yes: **Giovanni Scarzello

---

## [Author Response · Author response to Decision Letter 0]

2 Jun 2023

Reviewer #1

Comment 1 – I have read with great interest this systematic review about the role of elec-trochemotherapy for skin malignancies, in comparison with radiotherapy. The paper is well written and data are properly collected and presented. Correctly, the authors highlight the paucity of literature about the potential role of electrochemotherapy with very few experiences available. Nonetheless, I think that this article may help clinicians in the knowledge of this therapeutic op-tion.

We thank the reviewer for their kind comments.

Comment 2 – I only suggest to mention, in the Introduction, among the radiotherapy experiences for skin malignancies:

-doi: 10.1007/s00403-021-02268-1.

-doi: 10.4081/rt.2017.6942

We thank the reviewer for their suggestion. We found the articles they suggested interesting and relevant to the experience of radiotherapy. We have updated the manuscript to include these ar-ticles (paragraph 1, page 4).

 

Reviewer #2

Comment 1 – In Table 1 it would be useful for treating clinicians to know if there were any report-ed prior treatments to electrochemotherapy/RT to understand if these have a role in resistant or progressive disease. It would also be helpful to know how responses were measured in these studies (RECIST, radiological, clinically etc).

We thank the reviewer for their suggestion. We have updated Table 1 (page 10) to include more comprehensive details of the two included studies including previous therapies and the method of tumour response assessment.

Comment 2 – The study by Amaral et al. reported on PFS outcomes with electrochemotherapy vs RT (line 253, page 13) - this should be included in the summary results table on page 12 as PFS is a clinically relevant outcome when deciding between treatment options

We thank the reviewer for their suggestion. We have updated Table 1 (page 10) to include more comprehensive details of the two included studies including the progression free survival outcome reported by Amaral et al.

Comment 3 – Similarly, cohort details of the 2 provided studies should be provided in the sum-mary table - as the fact that Robertson et al.'s study was limited to patients with epidermolysis bullosa was not mentioned until much later (line 258, page 13). This is relevant as these results may not be as readily generalisable to clinicians treating cutaneous SCC beyond this rare sub-group

We thank the reviewer for their suggestion. We have updated Table 1 (page 10) to include more comprehensive details of the two included studies including characteristics of the study popula-tions.

Comment 4 – As acknowledged in the discussion, page 15 paragraph 1, the two studies are far too heterogenous both in design and in outcome reporting to allow for meaningful conclusions of electrochemotherapy vs radiotherapy to be drawn. This is a major issue with the review paper overall as it is not informative enough to guide practice, and this should be emphasised more clearly in the Abstract and conclusions.

We thank the reviewer for their suggestion. We have updated the conclusions section of the ab-stract (paragraph 2, page 3) and in the review conclusion (paragraph 2, page 18) to highlight this issue more clearly.

Comment 5 – Paragraph 2, page 15 is a repetition from the Results, and does not need to be highlighted again

We thank the reviewer for their suggestion. We have removed this paragraph and updated the "studies excluded and reasons for exclusion” section to clarify we contacted the authors to obtain extractable data (paragraph 2, page 11). We have also included a short paragraph in the discus-sion to succinctly emphasise that though there is data on these therapies in the literature, we en-countered significant challenges in extracting comparative data in this review (paragraph 2, page 15 to paragraph 1, page 16).

Comment 6 – Paragraph 3, page 15 is a description of studies identified in the literature search and should be moved to the Methods

We thank the reviewer for their comments. It is our understanding that conventionally, the outcome of article screening, including the exclusion of studies and the reason for their exclusion, is a result of the study, and should be described within the results section of the manuscript. We have there-fore moved this paragraph in the results section (paragraph 5, page 11, to paragraph 1, page 12). If the editorial decision is that this is the wrong place, we will gladly change it. We have also includ-ed a short paragraph in the discussion to succinctly emphasise that though there is data on these therapies in the literature, we encountered significant challenges in extracting comparative data in this review (paragraph 2, page 15 to paragraph 1, page 16).

Comment 7 – There needs to be more text in the Discussion describing the role of electrochemo-therapy or radiotherapy in the current treatment landscape of cutaneous metastases/solid tu-mours - these are not the only treatment options for patients and by not mentioning alternatives (isolated limb perfusion/infusion, systemic medications including checkpoint inhibitors for cSCC/melanoma or HH inhibitors for BCC, intralesional therapies such as T-VEC or IL-2 etc) the manuscript implies clinicians are limited to these two options

We thank the reviewer for their suggestion. Though the primary aim of this review was to compare electrochemotherapy with radiotherapy, we have included further detail in the discussion to high-lighted the existence of other available therapies (paragraph 2, page 17).

Comment 8 – The practical benefits for patients in comparison to these alternatives should be discussed in more detail - e.g. reduced time/financial toxicities, reduced systemic side effects etc

We thank the reviewer for their suggestion. Though the primary aim of this review was to compare electrochemotherapy with radiotherapy we have included additional detail within the discussion to examine some of the advantages and disadvantages of electrochemotherpay and radiotherapy compared with other treatment modalities (paragraph 2, page 17).

Comment 9 – All in all, I do not think the narrow range of results and conclusions that can be drawn from this study justify publication as such as lengthy article. It would be more suitable to shorten it to a Letter or Correspondence format; or to expand upon the search strategy to in-clude single-arm studies of either electrochemotherapy or radiotherapy in cutaneous metastases to provide a more meaningful indirect comparison (given that the two provided studies are both non-randomised and heterogenous themselves).

We thank the reviewer for their comment. We have taken great care to outline the paucity of evi-dence on this topic, and to report this in a succinct fashion. PLOS has no submission type for let-ter/short correspondence, and invites articles of any length. Shortening the manuscript would not allow for discussion of other treatment modalities as requested in earlier comments. We have re-vised the manuscript to remove unnecessary detail but maintain that this review draws attention to the lack of comparative data for these two therapies and provides a justification to clinicians to undertake further research in the form of clinical trials and observational studies. We align our-selves with the open science best practice of publication of negative findings, and also maintain this review provides a robust framework for a high quality systematic review of this topic which can be repeated in the future when further comparative data is published.

 

Reviewer #3

Comment 1 – Paper clear and easily understood even by the non-expert. Criteria for literature selection are thoroughly described. The conclusions are interlocutory, but this is due to lack of literature. I think it can be published.

We thank the reviewer for their kind comments.

---

## [Decision Letter · Decision Letter 1]

22 Jun 2023

Electrochemotherapy vs radiotherapy in the treatment of primary cutaneous malignancies or cutaneous metastases from primary solid organ malignancies: A systematic review and narrative synthesis

PONE-D-22-32656R1

Dear Dr. Totty,

We’re pleased to inform you that your manuscript has been judged scientifically suitable for publication and will be formally accepted for publication once it meets all outstanding technical requirements.

Kind regards,

Huijuan Cao, Ph.D.

Academic Editor

PLOS ONE

Additional Editor Comments (optional):

Reviewers' comments:

Reviewer's Responses to Questions

**Comments to the Author**

1. If the authors have adequately addressed your comments raised in a previous round of review and you feel that this manuscript is now acceptable for publication, you may indicate that here to bypass the “Comments to the Author” section, enter your conflict of interest statement in the “Confidential to Editor” section, and submit your "Accept" recommendation.

Reviewer #1: All comments have been addressed

Reviewer #2: All comments have been addressed

Reviewer #3: All comments have been addressed

2. Is the manuscript technically sound, and do the data support the conclusions?

Reviewer #1: Yes

Reviewer #2: Yes

Reviewer #3: Yes

3. Has the statistical analysis been performed appropriately and rigorously? 

Reviewer #1: Yes

Reviewer #2: N/A

Reviewer #3: Yes

4. Have the authors made all data underlying the findings in their manuscript fully available?

Reviewer #1: Yes

Reviewer #2: Yes

Reviewer #3: Yes

5. Is the manuscript presented in an intelligible fashion and written in standard English?

Reviewer #1: Yes

Reviewer #2: Yes

Reviewer #3: Yes

6. Review Comments to the Author

Reviewer #1: (No Response)

Reviewer #2: Thank you for your thorough revision to incorporate my suggestions. I believe the article reads much more clearly with more clinically useful datapoints emphasised more succinctly. Table 1 in particularly is a much more useful comparison of the included studies. The paragraph discussing the role of ECT and RT in the current treatment landscape of cutaneous malignancies/metastases is excellent. I am happy to recommend this revision for publication.

Reviewer #3: I thought the paper could be published even in its first draft; all the more so after this revision.

7. PLOS authors have the option to publish the peer review history of their article (what does this mean?). If published, this will include your full peer review and any attached files.

Reviewer #1: No

Reviewer #2: No

Reviewer #3: **Yes: **Giovanni Scarzello

---

## [Editor Report · Acceptance letter]

5 Jul 2023

PONE-D-22-32656R1 

Electrochemotherapy vs radiotherapy in the treatment of primary cutaneous malignancies or cutaneous metastases from primary solid organ malignancies: A systematic review and narrative synthesis 

Dear Dr. Totty:

I'm pleased to inform you that your manuscript has been deemed suitable for publication in PLOS ONE. Congratulations! Your manuscript is now with our production department. 

Kind regards, 

on behalf of

Dr. Huijuan Cao 

Academic Editor

PLOS ONE